# Reformulating Strict Monotonic Probabilities with a Generative Cost Model

## Abstract

In numerous machine learning contexts, the relationship between input variables and predicted outputs is not only statistically significant but also strictly monotonic. Conventional approaches to ensuring monotonicity focus primarily on construction or regularization methods. This paper establishes that the problem of strict monotonic probability can be interpreted as a comparison between an observable revenue variable and a latent cost variable. This insight allows us to reformulate the original monotonicity challenge into modeling the latent cost variable and estimating its distribution. To address this issue, we introduce a generative model for the latent cost variable, called the Generative Cost Model (**GCM**), and derive a corresponding loss function. We further enhance the estimation of latent variables using variational inference, which reformulate our loss function accordingly. Lastly, we validate our approach through a numerical simulation of quantile regression and several experiments on public datasets, demonstrating that our method significantly outperforms traditional techniques. The code of GCM is available in https://github.com/iclr-2025-4464/GCM.

## 1 Introduction

Many machine learning problems exhibit a monotonic relationship between inputs and outputs. Some of these relationships are statistical in nature, such as the correlation between a person's height and weight or the relationship between a company's stock price and its annual income. However, these monotonicities are often empirical and not strictly defined. In contrast, certain problems necessitate strict monotonicity, such as the relationship between equipment availability and its age, or the connection between auction winning rates and bidding prices. For these strict monotonic problems, we require a model capable of predicting strict monotonic probability based on specific input variables. We refer to these input variables as **revenue variables**, where higher revenue correlates with an increased probability of a more positive response.

The most common deep learning methods for addressing the monotonicity problem can be broadly categorized into two types (Runje & Shankaranarayana (2023)): monotonic by **construction** and by **regularization**. The construction approach maintains strict monotonicity through customized structures in deep neural networks, such as monotonic activation functions, positive weight matrices, and min-max structures (Sill (1997)). In contrast, the regularization approach promotes monotonicity by designing specific loss functions (Sill & Abu-Mostafa (1996)).

Unlike traditional approaches, we propose a novel method to tackle the monotonicity problem using a **generative** framework. In the estimation of $p(y|\boldsymbol{x}, \boldsymbol{r})$, where $y \in \mathbb{R}$ is a response that maintains monotonicity with respect to the revenue variable $\boldsymbol{r}$ but is not necessarily monotonic with respect to $\boldsymbol{x}$, we employ a two-step process. (i) We simplify the multivariate problem into a Bernoulli case via variable substitution trick, so that $y$ is reduced to binary values (0 or 1). (ii) We reformulate the monotonicity problem by defining a latent **cost variable** $\boldsymbol{c}$, such that $y = \mathbb{I}(\boldsymbol{c} \prec \boldsymbol{r}) \in \{0, 1\}$. This ensures that the monotonicity between $y$ and $\boldsymbol{r}$ is preserved, as we have $Pr(y = 1|\boldsymbol{x}, \boldsymbol{r}) = Pr(\boldsymbol{c} \prec \boldsymbol{r}|\boldsymbol{x}, \boldsymbol{r})$. Here, $\prec$ denotes the partial order in the vector space and $\mathbb{I}$ represents the indicator function. Through this transformation, we can bypass the need to design a strictly monotonic function and instead focus on the latent cost variable $\boldsymbol{c}$. Consequently, we can use any structure to model $\boldsymbol{c}$ with the monotonicity constraints being ignored, as the monotonicity is inherently satisfied by the definition of $\boldsymbol{c}$.

To generate the latent cost variable, we propose a two-stage generative process: (i) Sampling from joint prior: In the first stage, we sample three variables $x$, $r$ and $z$ from a joint prior $p_\theta(x, r, z)$. Here, $x$, $r$ are observable variables, while $z$ is a latent variable. We assume conditional independence holds: $z \perp\!\!\!\perp r \mid x$. This leads to the factorization of the joint distribution as $p_\theta(x, r, z) = p_\theta(z|x)p(x, r)$. (ii) Generating the cost variable: We generate the cost variable $c$ conditioned on $z$ using $p_\theta(c|z)$. This results in the joint distribution: $p_\theta(x, r, z, c) = p_\theta(c|z)p_\theta(z|x)p_\theta(x, r)$. Since we have generated $c$, by the definition $y = \mathbb{I}(c \prec r)$, we can express the evidence as: $p_\theta(x, r, y) = \int \int p_\theta(c|z)p_\theta(z|x)p(x, r)\mathbb{I}(c \vee_y r)dzdc$, where $\vee_y$ denotes $\prec$ if $y = 1$, and $\not\prec$ if $y = 0$ (note that $\not\prec$ is not equivalent to $\succeq$ in vector space). To simplify the model, we drop the term $p(x, z)$ and restrict our estimate of evidence to the conditional density $p_\theta(y|x, r) = \int \int p_\theta(c|z)p_\theta(z|x)\mathbb{I}(c\vee_y r)dzdc$, since $x$ and $r$ are always provided and we do not need to generate the entire evidence from scratch. Given that the latent variable $z$ is high-dimensional, accurately calculating the evidence requires integration over $z$, which can be computationally intensive. To address this, we propose two approaches to estimate the evidence: (i) Monte Carlo sampling on $z \sim p_\theta(z|x)$ to estimate $p_\theta(y|x, r)$. (ii) Use variational inference to obtain a lower bound on the evidence, which allows us to optimize the log-evidence by sampling $z$ from the recognition model $q_\phi(z|x, r, y)$.

In the last part, we conduct two types of experiments. First, we design a numerical simulation of the quantile regression task in which the predicted $r$th quantile increases monotonically with respect to the value of $r$. We compare the performance between conventional methods and our generative cost model. The results demonstrate that our method achieves superior predictive accuracy while preserving strict monotonicity. To further assess the performance of the multivariate revenue variable $r$, we conduct experiments on four public datasets: the Adult dataset (Becker & Kohavi (1996)), the COMPAS dataset (Larson et al. (2016)), the Diabetes dataset (Teboul) and the Blog Feedback dataset (Buza (2014)). In all four experiments, our model outperforms existing approaches. We perform several ablation studies to examine the impact of the hyperparameters in our generative cost model, with detailed findings provided in the Appendix C. In Appendix A, we design a card gamble simulation, proving that the predicted distribution of the latent cost variable $p_\theta(c|x)$ is converging towards the actual cost distribution.

The main contributions of our paper are summarized as follows:

- We introduce a universal technique that reformulates the problem of monotonic probability into a modeling problem for latent cost variables, avoiding restrictions in conventional monotonic neural networks.

- We address the modeling of the cost variable using a generative approach called the Generative Cost Model (GCM), and we present two loss functions derived from log-likelihood and the variational lower bound.

- We evaluate our method for classification tasks using a simulated quantile regression and tasks on four public datasets, demonstrating that our model consistently outperforms traditional monotonic models.

## 2 BACKGROUND

**Partial Order between Vectors.** For vectors $v_1$ and $v_2$ in $\mathbb{R}^n$, we define the partial order between $v_1$ and $v_2$ as: $v_1 \preceq v_2$ *if and only if* $v_1^{(k)} \leq v_2^{(k)}$, *for any* $1 \leq k \leq n$. This relationship is illustrated in Figure 1a. Note that $v_1 \preceq v_2$ is equivalent to $v_2 \succeq v_1$.

The strict order is defined by: $v_1 \prec v_2$ *if and only if* $v_1 \preceq v_2$ *and* $v_1 \neq v_2$. We have $v_1 \prec v_2$ is equivalent to $v_2 \succ v_1$, but not equivalent to $v_1 \not\succeq v_2$.

**Partial Order between Random Variables.** In this paper, we adopt the definition of first-order stochastic dominance (Hadar & Russell (1969)): *for random variables $r_1$ and $r_2$ defined on $\mathbb{R}^n$, we say that $r_2$ first-order stochastically dominates $r_1$ (denoted $r_1 \prec_1 r_2$) if and only if $Pr(r_1 \succ t) < Pr(r_2 \succ t)$ for any $t \in \mathbb{R}^n$.* Specifically, for one dimensional random variables, $r_1 \prec_1 r_2$ is equivalent to $F_1(t) > F_2(t)$ (or epi$F_1(t) \subset$ epi$F_2(t)$) for any $t \in \mathbb{R}$. where $F_i$ represents the cumulative distribution function (CDF) of the random variable $r_i$ and epi$F_i$ refers to the epigraph of the CDF.

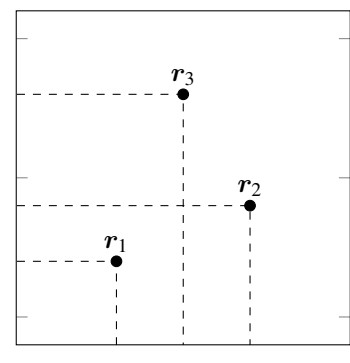
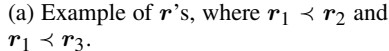
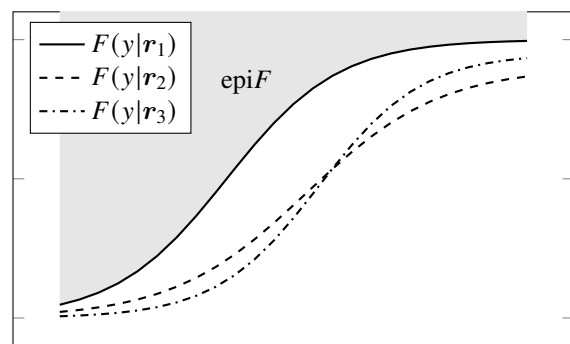

(a) Example of $r$'s, where $r_1 \prec r_2$ and $r_1 \prec r_3$.

(b) $r_1 \prec r_2 \Rightarrow \text{epi}F(y|r_1) \subset \text{epi}F(y|r_2)$ and $r_1 \prec r_3 \Rightarrow \text{epi}F(y|r_1) \subset \text{epi}F(y|r_3)$ due to the monotonicity of $y$ and $r$.

Figure 1: The CDFs of $F(y|r)$ with different $r$'s, where $y$ is monotonic with respect to $r$.

**Monotonic Conditional Probability.** *A conditional probability $p(y|r)$ is defined as monotonic, if and only if $y|r_1 \prec_1 y|r_2$ for any $r_1 \prec r_2$.* Or, in other words, $Pr(y \succ t|r_1) < Pr(y \succ t|r_2)$ for any vector $t$ and any pair $r_1 \prec r_2$. In this paper, we refer to the relationship between $y$ and $r$ as: $y$ being (conditionally) monotonic (increasing) with respect to $r$. All instances of monotonicity discussed here are assumed to be monotonically increasing; for decreasing relationships, we can simply replace the original variables with their opposites.

For example, if $y \sim \mathcal{N}(y; \mu, \Sigma)$, where the mean $\mu$ is also a random variable, then we find that $y$ is monotonic with respect to $\mu$. Similarly, if $y \sim \mathcal{Bernoulli}(\beta)$, then $y$ is monotonic with respect to $\beta$. In these cases, $\mu$ and $\beta$ are referred to as monotonic parameters of $y$.

The relationship between $r$ and $p(y|r)$ is illustrated in Figure 1, where $y$ is one-dimensional and monotonic with respect to $r$. In Figure 1a, we plot three random variables $r_1$, $r_2$ and $r_3$, with $r_1 \prec r_2$ and $r_1 \prec r_3$, while $r_2$ and $r_3$ are not comparable. Let $F(y|r_i)$ denote the CDF of $y$ conditioned on $r = r_i$. The corresponding conditional CDFs are plotted in Figure 1b, where $F(y|r_1)$ is positioned at the top with the smallest epigraph, while $F(y|r_2)$ intersects $F(y|r_3)$ indicating the incomparability between $r_2$ and $r_3$.

## 3 RELATED WORK

**Monotonic Modeling.** In many machine learning tasks, we have the prior knowledge that the output should be monotonic with respect to certain input variables. A straightforward idea is to identify a monotonic function and optimize its parameters to approximate the desired monotonic output. It can be summarized as the following form:

$$\begin{aligned} \text{minimize} \quad & \mathcal{L}(y, F_\theta(x, r)) \\ \text{subject to} \quad & \frac{\partial F_\theta(x, r)}{\partial r} \succ 0. \end{aligned} \tag{1}$$

The Min-Max architecture (Sill (1997)) is a pioneering work in monotonic neural networks, utilizing a piecewise linear model to approximate monotonic target functions. Its monotonicity is ensured through (i) positive weighting matrices, (ii) monotonic activation functions, and (iii) a Min-Max structure.

Along the direction of monotonic by construction, Nolte et al. (2022) introduced the Lipschitz monotonic network, which enhances robustness through weight constraint. Igel (2023) proposed the smoothed min-max monotonic network, which replaces the traditional min-max structure with a smoothed log-sum-exp function, preventing the network from becoming silent. Additionally, Runje & Shankaranarayana (2023) developed the constrained monotonic neural network, which improves the approximation of non-convex functions by modifying activation functions.

Another popular direction for improving monotonicity involves the use of regularization techniques, which can be formulated as:

$$\text{minimize} \quad \mathcal{L}(y, F_\theta(\boldsymbol{x}, \boldsymbol{r})) + \mathcal{R}(F_\theta), \tag{2}$$

where the regularization $\mathcal{R}(F_\theta) > 0$ if $F_\theta$ is not monotonic at some points. This direction includes monotonicity hints proposed by Sill & Abu-Mostafa (1996), which use hint samples and pairwise loss to guide model learning. The certified monotonic neural networks proposed by Liu et al. (2020) certify monotonicity by verifying the lower bound of the partial derivative of monotonic features. Furthermore, Gupta et al. (2019) proposed a pointwise penalization method for negative gradients, while counter example guided methods were introduced by Sivaraman et al. (2020).

In addition, the lattice networks (Garcia & Gupta (2009)) can solve the monotonic problem by either a construction or regularization approach; extensive works have been conducted in this area by Milani Fard et al. (2016), You et al. (2017), Gupta et al. (2019) and Yanagisawa et al. (2022), etc.

Monotonicity also plays an important role in many areas of machine learning. Ben-David (1995); Lee et al. (2003); van de Kamp et al. (2009); Chen & Guestrin (2016) bring monotonicity into tree models; Rashid et al. (2020) propose the QMIX method using monotonic value functions in multi-agent reinforcement learning; Lam et al. (2023) propose a multi-class loss function using monotonicity of gradients of convex functions; Haldar et al. (2020) and Xu et al. (2024) bring monotonicity into online business, etc.

**Variational Inference and Generative models.** Variational inference (VI) (Peterson (1987); Parisi & Shankar (1988); Saul & Jordan (1995)) is a powerful technique for working with generative models, and recent years have seen significant advances based on this approach (Kingma (2013); Rezende et al. (2014); Ozair & Bengio (2014); Burda et al. (2015); Sohl-Dickstein et al. (2015); Ho et al. (2020); Song et al. (2020)). VI transforms the complex task of Bayesian inference into a computationally manageable optimization problem by approximating the latent variables within a specified family of distributions. This is achieved by optimizing the evidence lower bound (ELB) rather than the original evidence.

Recent studies have highlighted the rapid growth of conditional generative models. In the realm of text-to-image generation, notable works include Ramesh et al. (2021), Ramesh et al. (2022), Saharia et al. (2022), and Rombach et al. (2022). For text-to-video generation, key contributions come from Esser et al. (2023) and Brooks et al. (2024). Unlike variational autoencoders (VAEs) (Kingma (2013)), which initiate generation from a latent variable, these conditional generative models begin with a pair comprising a given condition (such as text, image, or video) and a latent variable. This is typically expressed through the decomposition: $p(x, z) = p(x)p(z|x)$, where $x$ is the condition and $z$ is the latent variable. Consequently, these models primarily focus on conditional probability $p(z|x)$. In this paper, we adopt this paradigm to construct our cost generation model.

Moreover, the normalizing flow is an important subject of generative models, it not only transforms a simple distribution to a complicated distribution, but also requires these transformations to be invertible, which is sufficient when the transformations are continuous and monotonic. There have been studies that involve monotonicity in normalizing flows: Ziegler & Rush (2019); Ho et al. (2019); Wehenkel & Louppe (2019); Müller et al. (2019); Jaini et al. (2019); Dinh et al. (2019); Ahn et al. (2022).

## 4 THE COST VARIABLE METHOD

### 4.1 PROBLEM FORMULATION

Consider a binary classification problem of $(\boldsymbol{x}, \boldsymbol{r}, y)$, where $\boldsymbol{x} \in \mathbb{R}^n$ represents the ordinary variables, $\boldsymbol{r} \in \mathbb{R}^m$ is the revenue variable, and $y \in \{0, 1\}$ is the binary output variable that exhibits monotonicity with respect to $\boldsymbol{r}$. We assume that $y$ follows a Bernoulli distribution, with its mean parameter generated by a deep neural network $G : \mathbb{R}^n \times \mathbb{R}^m \to (0, 1)$:

$$y|\{\boldsymbol{x}, \boldsymbol{r}\} \sim \mathcal{B}ernoulli(y; G(\boldsymbol{x}, \boldsymbol{r})). \tag{3}$$

As defined in Section 2, the function $G$ has to be monotonic with respect to $\boldsymbol{r}$. We refer to $\boldsymbol{r}$ as the **revenue variable** associated with $y$. The rationale is that, when $y$ is viewed as a decision variable,

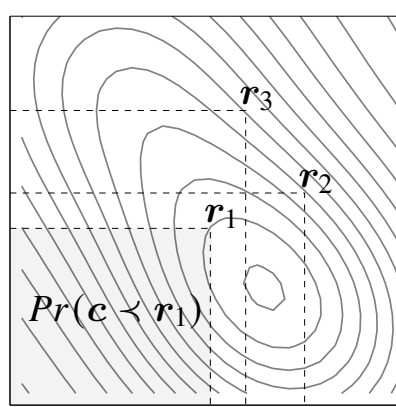

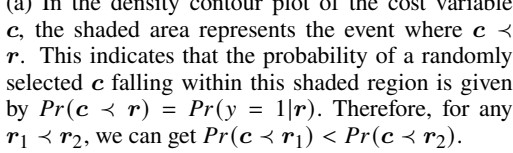

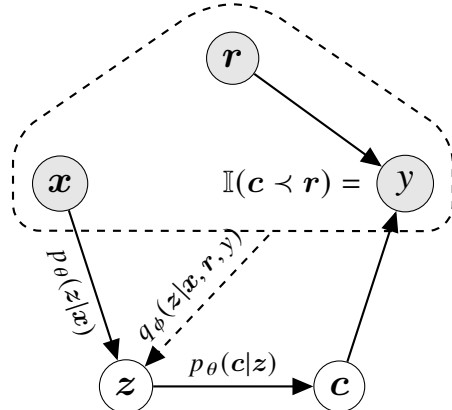

(a) In the density contour plot of the cost variable $c$, the shaded area represents the event where $c \prec r$. This indicates that the probability of a randomly selected $c$ falling within this shaded region is given by $Pr(c \prec r) = Pr(y = 1|r)$. Therefore, for any $r_1 \prec r_2$, we can get $Pr(c \prec r_1) < Pr(c \prec r_2)$.

(b) The graph illustrates the probability graphical model for a monotonic probability $p(y|x, r)$. In this model, the grey nodes represent observable variables $x$, $y$ and $r$, while the white nodes denote latent variables. Solid arrows indicate the generative model $p_\theta$, whereas the dashed arrow represents the recognition model $q_\phi$.

Figure 2: Definition (Figure 2a) and modeling (Figure 2b) of the latent cost variable.

a profit-maximizing decision will favor higher values of $r$, thus ensuring the monotonicity of $y$ with respect to $r$.

For a general monotonic problem of $(x, r, y)$ with continuous output $y \in \mathbb{R}$, the model takes the following form:

$$y|\{x, r\} \sim \mathcal{F}(y; G(x, r)), \tag{4}$$

where $\mathcal{F}$ denotes the chosen probability family for $y$. The function $G$ produces a monotonic parameter for $\mathcal{F}$ and is monotonic with respect to $r$. Consequently, $y$ maintains monotonicity with respect to $r$. For example, if $\mathcal{F}$ is a Gaussian distribution $\mathcal{N}(y; \mu(x, r), \sigma(x)^2)$ and $G = \mu(x, r)$ predicts its mean parameter, then $G$ must be a monotonic function of $r$ to ensure that $y$ is monotonic with respect to $r$.

To reduce the general monotonic probability problem to the binary scenario, we introduce an assistant random variable $t \in \mathbb{R}$ such that $t \perp\!\!\!\perp r \mid x$. We define the new response variable as $y^* = \mathbb{I}(y + t > 0) \in \{0, 1\}$ and the new revenue variable as $r^* = [t, r]$. For any $r_1^* \prec r_2^*$, since the monotonicity between $y$ and $r$, we have:

$$Pr(y^* = 1|r_1^*) = Pr(y > -t|r_1, t) < Pr(y > -t|r_2, t) = Pr(y^* = 1|r_2^*), \tag{5}$$

meaning $y^*$ is strictly monotonic with respect to $r^*$. In the opposite direction, if $y^*$ is monotonic with respect to $r^* = [t, r]$, then for any $r_1^* \prec r_2^*$ and $s \in \mathbb{R}$, we have $[-s, r_1] \prec [-s, r_2]$ and $Pr(y > s|r_1) < Pr(y > s|r_2)$, proving that $y|r_2 \succ y|r_1$. This establishes the equivalence between the problems of $(y^*, x, r^*)$ and $(y, x, r)$. Therefore, the monotonic modeling problem of the triplet $(y, x, r)$ where $y \in \mathbb{R}$ is reduced to the binary problem of $(y^*, x, r^*)$, which is $y + t > 0|\{x, r\} \sim \mathcal{B}ernoulli(y^*; G(x, t, r]))$. Since $Pr(y \leq s|x, r^*) = 1 - Pr(y > s|x, r^*) = 1 - G(x, [r, -s])$, the density function of $y$ is

$$p(y|x, r) = -\frac{\partial G(x, [r, -s])}{\partial s}\bigg|_{s=y}. \tag{6}$$

Which completes the transformation from a general monotonic probability problem to a binary monotonic problem. We give an example of calculating the maximum likelihood estimate of $y$ as well as deriving the MLE loss function in Appendix B.3.

## 4.2 MONOTONICITY VIA THE COST VARIABLE

We now focus on the binary problem. The traditional approach, as defined in Equation 3, involves identifying a strictly (or weak) monotonic function $G(\boldsymbol{x}, \boldsymbol{r})$ with respect to $\boldsymbol{r}$. In this paper, instead of searching for a suitable function $G$, we introduce a random variable $\boldsymbol{c}$ to model $y$ defined by:

$$y = \mathbb{I}(\boldsymbol{c} \prec \boldsymbol{r}). \tag{7}$$

Given that $\{\boldsymbol{c}|\boldsymbol{c} \prec \boldsymbol{r}_1\} \subset \{\boldsymbol{c}|\boldsymbol{c} \prec \boldsymbol{r}_2\}$, for any $\boldsymbol{r}_1 \prec \boldsymbol{r}_2$, it follows that $Pr(y = 1|\boldsymbol{r} = \boldsymbol{r}_1) < Pr(y = 1|\boldsymbol{r} = \boldsymbol{r}_2)$, which guarantees that $y$ is strictly monotonic with respect to $\boldsymbol{r}$. Then we can define:

$$G(\boldsymbol{x}, \boldsymbol{r}) = \mathbb{E}[y|\boldsymbol{x}, \boldsymbol{r}] = Pr(\boldsymbol{c} \prec \boldsymbol{r}|\boldsymbol{x}, \boldsymbol{r}) = \int_{\boldsymbol{c} \prec \boldsymbol{r}} p(\boldsymbol{c}|\boldsymbol{x}) d\boldsymbol{c}, \tag{8}$$

demonstrating that $y|\{\boldsymbol{x}, \boldsymbol{r}\} \sim \mathcal{Bernoulli}(G(\boldsymbol{x}, \boldsymbol{r}))$. Thus, $G(\boldsymbol{x}, \boldsymbol{r})$ serves as the monotonic function proposed in Equation 3 . Notably, we do not need to derive the exact form of $G$, as long as we can estimate the conditional density $p(\boldsymbol{c}|\boldsymbol{x})$.

Unlike conventional methods that require $G$ to be a strictly monotonic function, there are no constraints on $p(\boldsymbol{c}|\boldsymbol{x})$. We can take any form of $p(\boldsymbol{c}|\boldsymbol{x})$, and the monotonicity of $p(y|\boldsymbol{r})$ holds strictly due to the definition of $y$ in Equation 7. We call $\boldsymbol{c}$ the **cost variable**. As illustrated in Figure 2a, the probability of $y$ is equivalent to the probability that the revenue $\boldsymbol{r}$ domains the cost $\boldsymbol{c}$, that is, $Pr(y = 1) = Pr(\boldsymbol{c} \prec \boldsymbol{r})$. Thus, the original task of finding a monotonic function $G$ reduces to determining the distribution of $\boldsymbol{c}$. However, since $\boldsymbol{c}$ is a latent variable, we must infer $\boldsymbol{c}$ based on the observable variables $\boldsymbol{x}$, $\boldsymbol{r}$ and $y$, which is a challenge that still needs to be addressed.

## 4.3 GENERATIVE COST MODEL

As we focusing on modeling the cost variable $\boldsymbol{c}$, the distribution of $\boldsymbol{c}$ can be complicated, making it challenging to select an appropriate distribution family. To bypass the need for choosing a suitable distribution family, we adopt a generative approach that can automatically approximate complicated distributions. In this paper, we construct a simple generative model for $\boldsymbol{c}$ through the following process:

$$\begin{aligned}
\boldsymbol{x}, \boldsymbol{r} &\sim p(\boldsymbol{x}, \boldsymbol{r}), \\
\lambda_z = \text{DNN}_z(\boldsymbol{x}; \theta_1), \ p_{\theta_1}(\boldsymbol{z}|\boldsymbol{x}) &= \mathcal{P}_z(\boldsymbol{z}; \lambda_z), \\
\lambda_c = \text{DNN}_c(\boldsymbol{z}; \theta_2), \ p_{\theta_2}(\boldsymbol{c}|\boldsymbol{z}) &= \mathcal{P}_c(\boldsymbol{c}; \lambda_c), \\
y &= \mathbb{I}(\boldsymbol{c} \preceq \boldsymbol{r}).
\end{aligned} \tag{9}$$

The generative model consists of three independent stages: $p(\boldsymbol{x}, \boldsymbol{r})$, $p_{\theta_1}(\boldsymbol{z}|\boldsymbol{x})$ and $p_{\theta_2}(\boldsymbol{c}|\boldsymbol{z})$, where $\theta = [\theta_1, \theta_2]$ are the generative parameters that must be learned. We do not need to model the first stage since $\boldsymbol{x}$ and $\boldsymbol{r}$ are always given during inference. In the second stage, we generate the latent variable $\boldsymbol{z}$ via $p_{\theta_1}(\boldsymbol{z}|\boldsymbol{x})$. Subsequently, the latent cost variable $\boldsymbol{c}$ is generated by $p_{\theta_2}(\boldsymbol{c}|\boldsymbol{z})$ , which is set to be elementwise independent, that gives us the decomposition

$$p_{\theta_2}(y|\boldsymbol{z}, \boldsymbol{r}) = p_{\theta_2}(\boldsymbol{c} \vee_y \boldsymbol{r}|\boldsymbol{z}, \boldsymbol{r}) = 1 - y - (-1)^y \prod_i \int_{-\infty}^{\boldsymbol{r}^{(i)}} p_{\theta_2}(\boldsymbol{c}^{(i)}|\boldsymbol{z}) d\boldsymbol{c}^{(i)}. \tag{10}$$

As illustrated in Figure 2b, we assume that the conditional independencies: $\boldsymbol{z} \perp\!\!\!\perp \boldsymbol{r} \mid \boldsymbol{x}$ and $\boldsymbol{x} \perp\!\!\!\perp y \mid \{\boldsymbol{z}, \boldsymbol{r}\}$ hold (we discuss another assumption in Appendix D where we abandon $\boldsymbol{z} \perp\!\!\!\perp \boldsymbol{r} \mid \boldsymbol{x}$). Thus the probability of $y$ conditioned on $\boldsymbol{x}$ and $\boldsymbol{r}$ can be formulated as:

$$p_\theta(y|\boldsymbol{x}, \boldsymbol{r}) = \int p_{\theta_1}(\boldsymbol{z}|\boldsymbol{x}, \boldsymbol{r}) p_{\theta_2}(y|\boldsymbol{z}, \boldsymbol{x}, \boldsymbol{r}) d\boldsymbol{z} = \int p_{\theta_1}(\boldsymbol{z}|\boldsymbol{x}) p_{\theta_2}(y|\boldsymbol{z}, \boldsymbol{r}) d\boldsymbol{z} = \mathbb{E}_{\boldsymbol{z} \sim p_{\theta_1}} p_{\theta_2}(y|\boldsymbol{z}, \boldsymbol{r}). \tag{11}$$

To find the optimal parameter $\theta = [\theta_1, \theta_2]$, we maximize the log-likelihood ($LL$) of the observation $y$, which is:

$$\begin{aligned}
LL = \log p_\theta(y|\boldsymbol{x}, \boldsymbol{r}) &= \log \mathbb{E}_{\boldsymbol{z}_k \sim p_{\theta_1}(\boldsymbol{z}|\boldsymbol{x})} \left[ \frac{1}{K} \sum_{k=1}^K p_{\theta_2}(y|\boldsymbol{z}_k, \boldsymbol{r}) \right] \\
&\geq \mathbb{E}_{\boldsymbol{z}_k \sim p_{\theta_1}(\boldsymbol{z}|\boldsymbol{x})} \log \left[ \frac{1}{K} \sum_{k=1}^K p_{\theta_2}(y|\boldsymbol{z}_k, \boldsymbol{r}) \right].
\end{aligned} \tag{12}$$

To maximize $LL$, we can alternatively maximize the RHS of Equation 12, which can be estimated by sampling $z_k \sim p_{\theta_1}(z|x)$, $k = 1, \cdots, K$. Since we need to optimize both parameters $\theta_1$ and $\theta_2$ via gradient descent methods, we adopt the reparameterization trick (Kingma (2013)) as the following form:

$$z(\theta_1, x, r, \epsilon) = \mu_{\theta_1}(x, r) + \sigma_{\theta_1}(x, r) \odot \epsilon, \tag{13}$$

where $\epsilon \sim \mathcal{N}(0, E)$. Therefore, the final GCM loss function is:

$$\mathcal{L}_{GCM}(\theta; x, r, y) = -\log \frac{1}{K} \sum_{k=1}^{K} p_{\theta_2}(y|x, r, z(\theta_1, x, r, \epsilon_k)). \tag{14}$$

The details of the model is available in the Appendix B.1. However, when $z$ is a $K$-categorical variable that $z \in \{1, \cdots, K\}$, we have the exact estimate of $LL = \log \sum_{k=1}^{K} p_{\theta_1}(z = k|x) p_{\theta_2}(y|z = k, r)$. This avoids the uncertainty of sampling on $z \sim p_{\theta_1}(z|x)$, which is useful when the dimension of $r$ and $c$ is small enough that we do not need a complex latent variable $z$ to model the low-dimensional cost variable $c$. The details of our model with categorical $z$ are available in the Appendix B.2.

## 4.4 GENERATIVE COST MODEL WITH VARIATIONAL INFERENCE

A significant challenge arises from the difficulty in learning the distribution of $z$ conditioned on $x$ when the latent distribution is complex. To improve the modeling of $z$, we introduce the recognition model $q_\phi(z|x, r, y)$ that use all the observable variables to approximate the intractable posterior $p_\theta(z|x, r, y)$, the recognition model is formulated as:

$$\lambda_{\tilde{z}} = \text{DNN}_{\tilde{z}}(x, r, y; \phi), \ q_\phi(z|x, r, y) = \mathcal{P}_{\tilde{z}}(z; \lambda_{\tilde{z}}). \tag{15}$$

Similar to the IWAE (Burda et al. (2015)), by Jensen's inequality, we have the evidence lower bound (ELB):

$$ELB = \mathbb{E}_{z_k \sim q_\phi} \log \left[ \frac{1}{K} \sum_{k=1}^{K} \frac{p_\theta(y, z_k|x, r)}{q_\phi(z_k|x, r, y)} \right] \leq \log \mathbb{E}_{z_k \sim q_\phi} \left[ \frac{1}{K} \sum_{k=1}^{K} \frac{p_\theta(y, z_k|x, r)}{q_\phi(z_k|x, r, y)} \right] = \log p_\theta(y|x, r). \tag{16}$$

So the objective of the variational version of GCM (noted as GCM-VI) is:

$$\mathcal{L}_{GCM-VI}(\theta, \phi; x, r, y) = -\log \left[ \frac{1}{K} \sum_{k=1}^{K} \frac{p_{\theta_2}(y|z_k, r) p_{\theta_1}(z_k|x)}{q_\phi(z_k|x, r, y)} \right]. \tag{17}$$

Here, $z_k \sim q_\phi(z|x, r, y)$ is sampled through the reparameterization trick similar to Equation 13. Ablation studies for values of the latent dimension $D$ and the sample number $K$ are available in the Appendix C.

## 5 EXPERIMENT

### 5.1 EXPERIMENT OF QUANTILE REGRESSION BY SIMULATION

Quantile regression is a common problem in statistics, its goal is to estimate the $r$th quantile of $y$ conditioned on $x$, based on observations of $x$ and $y$. The $r$th quantile $Q_{y|x}(r)$ is defined by $Q_{y|x}(r) = F_{y|x}^{-1}(r)$, where $F_{y|x}$ is the conditional cumulative distribution function of $y$ conditioned on $x$. Since $F$ is monotonic, its inverse $Q_{y|x}(r)$ is also strict monotonic with respect to $r$. The common objective (Koenker (2005)) of the linear quantile regression is given by:

$$\hat{\beta}_r = \arg \min_{\beta_r} \sum_{i=1} (r(y^{(i)} - \hat{y}^{(i)})_+ + (1 - r)(\hat{y}^{(i)} - y^{(i)})_+), \tag{18}$$

where $\hat{y}_r^{(i)} = x^{(i)} \beta_r$ is a linear prediction of the quantile $Q_{y|x}(r)$ and $\beta_r$ is its parameter. For the nonlinear $y|x$, we can adopt neural networks to capture such relationship automatically. In addition, we can introduce $r$ into the network and predict the $r$th quantile of $y|x$ by $\hat{y}_r = \text{DNN}_\theta(x, r)$ for any $r \in (0, 1)$. Or, in a generative style:

$$y_r \sim p_\theta(y_r|x, r). \tag{19}$$

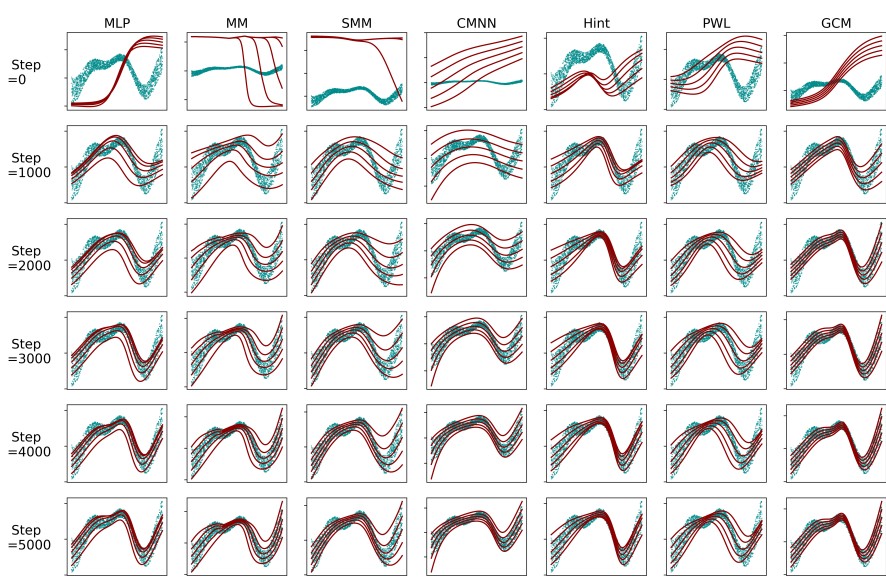

Figure 3: Plot of $\hat{y}_r|(x,r)$ for $r \in \{0.1, 0.3, 0.5, 0.7, 0.9\}$ (red curves). The background scatters are real training samples.

However, this problem is different from the original monotonic modeling, since the variable $r$ here is unobservable. To solve this issue, we modify the monotonic modeling problem into the following form:

$$
\begin{aligned}
&\text{sample } r \sim \mathcal{U}([0,1]) \\
&\text{sample } \hat{y}_r \sim p_\theta(y|x,r) \\
&\text{minimize } r(y-\hat{y}_r)_+ + (1-r)(\hat{y}_r - y)_+.
\end{aligned}
\tag{20}
$$

And now we can do experiments based on the typical monotonic methods. In this experiment, we compare several classic methods with our generative model (GCM), all of which share the same baseline architecture: a three-layer perceptron network with tanh activations. During training, we employ the classic stochastic gradient descent method to optimize network parameters.

The methods we compare include: (i) the baseline MLP network (MLP); (ii) Min-Max network (MM) (Sill (1997)); (iii) smoothed Min-Max network (SMM) (Igel (2023)); (iv) constrained monotonic network (CMNN); (v) monotonicity hint model (Hint) (Sill & Abu-Mostafa (1996)); (vi) pointwise loss method (PWL) (Gupta et al. (2019)). Note that the MLP method does not require monotonicity, it does not face the difficulties in strict monotonic structure designing as other methods. Here we regard it as a benchmark of a free-style model but not a baseline of the monotonic modeling family. The Hint and PWL methods are weak monotonic methods which encourage but do not assure strict monotonicity. The method to be tested is the GCM with a categorical latent variable $z$, following the same procedure as formulated in Appendix B.2 and Appendix B.3, and here we take the latent categorical dimension as 8.

The training data are generated through a simulation with the setting:

$$
y = 0.3\sin(2(x+0.8)) + 0.4\sin(3(x-1.3)) + 0.3\sin(5x) + 0.4(0.8x^2 + 0.6)\epsilon,
\tag{21}
$$

where $x \in (-1.5, 1.5)$ and $\epsilon \sim \mathcal{U}(0,1)$. For each sampled $(x,y)$, we additionally sample $r \sim \mathcal{U}([0,1])$ and optimize our models following Equation 20. Note that the sampling of $r$ is independent of the sampling of $\epsilon$. We train our model with batch size of 20 in 5,000 rounds, resulting in a total of 100,000 training examples, while the models are tested on 1,000 examples. The test results during the training process are shown in Figure 3, where the results are demonstrated by the $r$th

quantile curve $(x, \hat{y}_r)$ for $r \in \{0.1, 0.3, 0.5, 0.7, 0.9\}$. We can see that GCM predicts the most accurate quantile values of $\hat{y}_r$, as well as maintaining a strict monotonicity between $\hat{y}_r$ and $r$. The traditional strict monotonic methods (MM, SMM, CMNN) suffer from approximation accuracy, as the strict monotonic structures (e.g. positive weighting matrices and monotonic activations) weaken the universal approximation ability of neural networks. The non-monotonic (MLP) and weak-monotonic (Hint, PWL) methods have better approximation accuracy than the strict monotonic methods. However, for these methods, the curves of $\hat{y}_r$ with different $r$'s are not sufficiently separated, due to the lack of strict monotonic constraints.

In Table 1, we show the detailed mean absolute error (MAE) of all methods in the quantile regression task, with $r \in \{0.1, 0.3, 0.5, 0.7, 0.9\}$. We repeat the experiment 10 times with different random seeds, and the final results are reported with 95% confidence intervals. As the data show, GCM performs the best among all the methods.

Table 1: MAE (with 95% confidence interval) of the quantile regression experiment.

| Method | MAE | | | | |
| | $r$=0.1 | $r$=0.3 | $r$=0.5 | $r$=0.7 | $r$=0.9 |
| --- | --- | --- | --- | --- | --- |
| MLP | 0.1495 ±0.0340 | 0.1157 ±0.0283 | 0.1057 ±0.0255 | 0.1230 ±0.0309 | 0.1477 ±0.0386 |
| MM | 0.2002 ±0.0572 | 0.1103 ±0.0320 | 0.0723 ±0.0245 | 0.1067 ±0.0346 | 0.1745 ±0.0495 |
| SMM | 0.2345 ±0.0693 | 0.1194 ±0.0356 | 0.0812 ±0.0246 | 0.1236 ±0.0366 | 0.1919 ±0.0556 |
| CMNN | 0.1768 ±0.0340 | 0.1119 ±0.0174 | 0.0823 ±0.0161 | 0.1007 ±0.0198 | 0.1480 ±0.0332 |
| Hint | 0.1402 ±0.0285 | 0.1137 ±0.0263 | 0.1068 ±0.0292 | 0.1154 ±0.0368 | 0.1316 ±0.0374 |
| PWL | 0.1793 ±0.0282 | 0.1476 ±0.0164 | 0.1394 ±0.0193 | 0.1524 ±0.0216 | 0.1698 ±0.0207 |
| GCM | **0.0984** ±0.0188 | **0.0777** ±0.0119 | **0.0669** ±0.0096 | **0.0759** ±0.0127 | **0.0991** ±0.0211 |

## 5.2 EXPERIMENTS FOR MULTIDIMENSIONAL REVENUE ON PUBLIC DATASETS

To further evaluate the GCM model for the multidimensional revenue variable, we use four public datasets: the Adult dataset (Becker & Kohavi (1996)), the COMPAS (Correctional Offender Management Profiling for Alternative Sanctions) dataset (Larson et al. (2016)), the Diabetes dataset (Teboul) and the Blog Feedback dataset (Buza (2014)). The property of each dataset is shown in Table 2 .

Table 2: Details of the datasets.

| dataset | total examples | dimension of $x$ | dimension of $r$ | target |
| --- | --- | --- | --- | --- |
| Adult | 48,842 | 33 | 4 | classification |
| COMPAS | 7,214 | 9 | 4 | classification |
| Diabetes | 253,680 | 105 | 4 | classification |
| Blog Feedback | 52,397 | 272 | 8 | regression |

The model we test are the same as we presented in Section 5.1, while the evaluation metrics are switched to log-loss, RMSE, AUC and ACC. And, as we stated in Section 5.1, we regard the MLP model as a benchmark of a freestyle model but not a baseline of the monotonic modeling family. For all four datasets, the training and testing sets are split in a 4:1 ratio. We also follow the data preprocessing procedures outlined by Liu et al. (2020) for the COMPAS dataset. For the Blog Feedback dataset, we perform a logarithm transformation for numerical features and target value. In all experiments, we employ the Gaussian distribution for latent $z$ in the GCM and GCM-VI, the hyperparameter settings of GCM and GCM-VI are $D = 4$ and $K = 32$. The testing results are demonstrated in Table 3 and the full results are available in Appendix E. All experiments are repeated 10 times with different random seeds, the final results are reported with a 95% confidence interval.

Our GCM and GCM-VI models achieve the top two performances in all metrics in all datasets after $10,000$ training steps. Notably, GCM-VI achieves the best performance on all datasets except the Blog Feedback dataset, proving the effectiveness of introducing variational bound into our generative

Table 3: Experimental results on the multiple datasets.

| Method | Adult ACC↑ | COMPAS ACC↑ | Diabetes ACC↑ | Blog Feedback RMSE↓ |
|---|---|---|---|---|
| MLP ∗ | 0.8837 ±0.0012 | 0.6955 ±0.0008 | 0.8431 ±0.0004 | 0.1042 ±0.0004 |
| MM | 0.8836 ±0.0010 | 0.6949 ±0.0021 | 0.8409 ±0.0008 | 0.1100 ±0.0018 |
| SMM | 0.8837 ±0.0011 | 0.6955 ±0.0020 | 0.8401 ±0.0013 | 0.1114 ±0.0008 |
| CMNN | 0.8832 ±0.0013 | 0.6997 ±0.0011 | 0.8393 ±0.0015 | 0.1118 ±0.0005 |
| Hint ♯ | 0.8846 ±0.0011 | 0.6861 ±0.0024 | 0.8407 ±0.0005 | 0.1118 ±0.0013 |
| PWL ♯ | 0.8835 ±0.0012 | 0.6960 ±0.0013 | 0.8417 ±0.0003 | 0.1069 ±0.0006 |
| GCM | 0.8854 ±0.0013 | 0.6991 ±0.0011 | 0.8441 ±0.0001 | **0.0994** ±0.0003 |
| GCM VI | **0.8858** ±0.0014 | **0.7011** ±0.0011 | **0.8442** ±0.0002 | 0.1005 ±0.0004 |

∗: No monotonicity requirements.
♯: Weak monotonicity via regularization.

objective. The detailed results are available in the Appendix E.1. And a time complexity analysis is available in the Appendix F.

## 6 CONCLUSION

This paper presents an innovative generative method for monotonic modeling by reformulating the monotonicity problem through the incorporation of a latent cost variable $c$. We have developed a robust generation process for this cost variable that accurately approximates the latent costs. Our experimental results demonstrate that the proposed Generative Cost Model (GCM and GCM-VI) effectively addresses the monotonicity challenge, significantly outperforming traditional approaches across various tasks.

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

## A  A GAMBLE SIMULATION

We design a card gamble and the rules are listed as follows:

- There are $n$ cards, each labeled with a number from 1 to $n$. The cards are shuffled and then the backsides are also labeled with numbers from 1 to $n$.
- In each round, the dealer shuffles the cards and then the player picks $l$ cards from the top of the deck. The player sees the front sides of the selected cards and places a bet of $r$ chips, where $r < n$.
- The dealer rolls a dice to select one card from the $l$ selected cards. If the backside number of this card is less than $r$, the player wins and receives $n$ chips as a prize, producing a net profit of $n - r$, otherwise the player loses, resulting in a profit of $-r$.
- In the whole game, a player can only see the front side, but not the backside of all cards.

In our gambling model, the rules state that the more chips a player bets, the higher the likelihood of winning, but correspondingly the prize of winning shrinks. We denote the winning event as $y = 1$ and the losing as $y = 0$. The selected cards of the player are represented as $\boldsymbol{x} = [\boldsymbol{x}_1, \cdots, \boldsymbol{x}_l]$, where each $\boldsymbol{x}_i$ corresponds to an embedding of the $i$ th card. Consequently, the probability $p(y|\boldsymbol{x}, r)$ is strictly monotonic with respect to the bet $r$. We train our generative cost model on a simulated dataset and evaluate the performance of our model $p_\theta(y|\boldsymbol{x}, r)$ using the same strategy. To assess the prize-winning capability of the models, we determine the optimal bet of model $p_\theta$ is: $r^* = \arg\max_r \{ p_\theta(y|\boldsymbol{x}, r)n - r. \}$ The real profit generated by the choice $r^*$ is $\mathbb{I}(r^* > c)n - r^*$. To maximize the total profit, a model has to learn the probability $p_\theta(y|\boldsymbol{x}, r)$ accurately for every combinations of $\boldsymbol{x}$ and $r$.

The cost variable $c$ corresponds to a random choice of the unobservable values on the backsides of the picked cards $\boldsymbol{x}_1, \cdots, \boldsymbol{x}_l$, and we note these backside values as $b_1, \cdots, b_l$. As a result, the model should infer the probabilities of the backside value of each $\boldsymbol{x}_i$. This inference is particularly challenging, as the models can only deduce these probabilities from training samples consisting of $(\boldsymbol{x}, r, y)$. In particular, the optimal solution for the generative cost model is to learn a precise mapping from $\boldsymbol{x}$ to $p(c|\boldsymbol{x})$, which is given by:

$$p(c|\boldsymbol{x}) = \frac{\mathbb{I}(c \in \{b_1, \cdots, b_l\})}{l} \tag{22}$$

We evaluate these methods using the following metrics: (i) the area under the precision-recall curve (AUC) between $p_\theta(y|\boldsymbol{x}, r)$ and $y$; (ii) the Kullback-Leibler (KL) divergence between $p_\theta(y|\boldsymbol{x}, r)$ and the true $p(y|\boldsymbol{x}, r)$; (iii) Kendall's $\tau$ coefficient, calculated between multiple pairs of $p_\theta(y|\boldsymbol{x}, r)$ and $r$ with fixed $\boldsymbol{x}$, for validating models' monotonicity; (iv) the prize money earned by each model.

In our experiments, we evaluate the two proposed methods: the Generative Cost Model (GCM). For the GCM, we utilize a categorical latent variable $\boldsymbol{z}$ and estimate the likelihood as demonstrated in the Appendix B.2. The model is trained on simulated data derived from the card game we designed, with hyperparameters set to $n = 10,000$ and $l = 4$. We assume that $r$ is generated independently of $x$. We train our model with mini-batches of size 100 in 50,000 rounds, resulting in a total of 5,000,000 training examples, while the methods are tested on 100,000 examples. The experimental results comparing our models with other methods are summarized in Table 4.

As shown in Table 4, our experiments demonstrate that the Generative Cost Model (GCM) achieves superior performance compared to all other monotonic methods. Notably, the performance on

Table 4: Experimental results (with a 95% confidence interval) for the simulated card game.

| Method | AUC↑ | KL Div.↓ | Kendall's $\tau$↑ | Prize Profit↑ |
|---|---|---|---|---|
| MLP | 0.8803 ±0.0006 | 0.0630 ±0.0012 | 0.8989 ±0.0042 | 1053.7 ±24.9 |
| MM | 0.8844 ±0.0012 | 0.0578 ±0.0033 | **1** ±0 | 1251.5 ±68.1 |
| SMM | 0.8824 ±0.0031 | 0.0629 ±0.0072 | **1** ±0 | 1104.6 ±130.6 |
| CMNN | 0.8823 ±0.0013 | 0.0624 ±0.0029 | **1** ±0 | 1025.1 ±35.0 |
| Hint | 0.8850 ±0.0013 | 0.0585 ±0.0028 | 0.9499 ±0.0027 | 1164.1 ±71.0 |
| PWL | 0.8879 ±0.0013 | 0.0526 ±0.0036 | **1** ±0 | 1355.9 ±91.4 |
| GCM | **0.8917** ±0.0005 | **0.0395** ±0.0019 | **1** ±0 | **1699.2** ±48.1 |

Kendall's $\tau$ coefficient meets our expectations, as these models ensure strict monotonicity; the only exceptions are the MLP model and the Hint model, which fail to predict monotonic results since their architecture do not assure strict monotonicity.

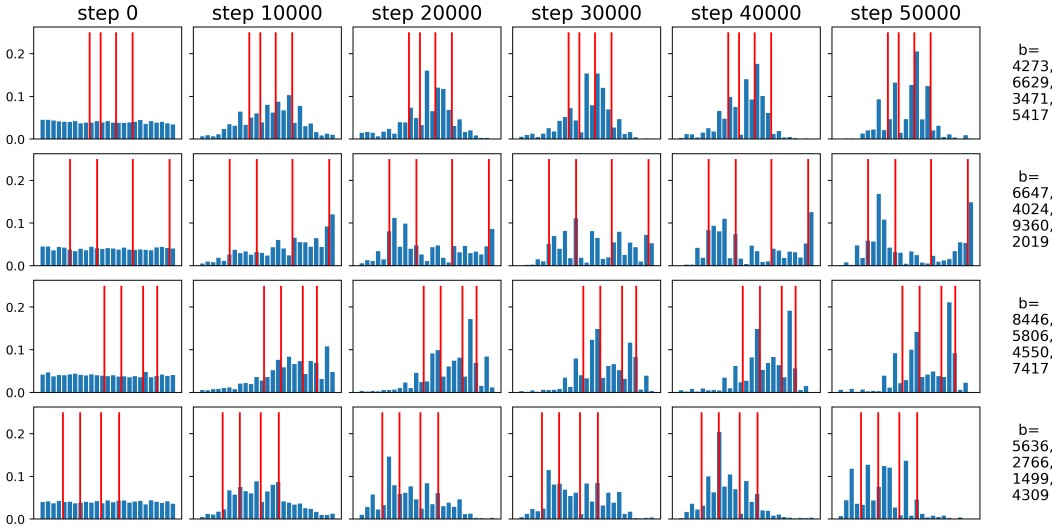

Figure 4: The predicted distribution of $p_\theta(c|\boldsymbol{x})$ (histogram in blue) by GCM is compared to the actual distribution of $c$ (represented by the red lines). In each row, we fix the variable $\boldsymbol{x}$ and the actual $p(c|\boldsymbol{x})$. As the training progresses, $p_\theta(c|\boldsymbol{x})$ gradually converges to $p(c|\boldsymbol{x})$.

Since our model focuses on modeling the distribution of the latent cost variable $c$, we can leverage the actual distribution of $c$ formulated in Equation 22. During the training process, we record the prediction of $p_\theta(c|\boldsymbol{x}) = \mathbb{E}_{z \sim p_\theta(z|\boldsymbol{x})}(c|z)$. As shown in Figure 4, the predicted density of $c$ is increasingly aligned with the actual distribution as training progresses. This observation confirms that our generative cost model effectively learns the latent cost variable.

# B  Details of GCM

## B.1  Gaussian Case

The generative model with Gaussian latent variable $z$ is designed by:

$$
\begin{aligned}
&\boldsymbol{\mu}, \log \boldsymbol{\sigma}^2 = \mathrm{DNN}_z(\boldsymbol{x}; \theta_1), \\
&\boldsymbol{\epsilon} \sim \mathcal{N}(\mathbf{0}, \boldsymbol{E}), \\
&\boldsymbol{z} = \boldsymbol{\mu} + \boldsymbol{\sigma} \odot \boldsymbol{\epsilon}, \\
&\boldsymbol{\mu}_c, \boldsymbol{s}_c = \mathrm{DNN}_c(\boldsymbol{z}; \theta_2), \\
&\boldsymbol{c} \sim \mathcal{Logistic}(\boldsymbol{\mu}_c, \boldsymbol{s}_c), \\
&Pr(\boldsymbol{c} \preceq \boldsymbol{r}) = \prod_i \mathrm{sigmoid}\left(\frac{\boldsymbol{r}^{(i)} - \boldsymbol{\mu}_c^{(i)}}{\boldsymbol{s}_c^{(i)}}\right).
\end{aligned}
\tag{23}
$$

For GCM-VI, the recognition encoder is:

$$
\begin{aligned}
&\hat{\boldsymbol{\mu}}, \log \hat{\boldsymbol{\sigma}}^2 = \mathrm{DNN}_{\hat{z}}(\boldsymbol{x}, \boldsymbol{r}, \boldsymbol{y}; \theta_3) \\
&\boldsymbol{\epsilon} \sim \mathcal{N}(\mathbf{0}, \boldsymbol{E}), \\
&\hat{\boldsymbol{z}} = \hat{\boldsymbol{\mu}} + \hat{\boldsymbol{\sigma}} \odot \boldsymbol{\epsilon},
\end{aligned}
\tag{24}
$$

while the decoder shares with GCM.

## B.2  Categorical Case

The generative model with categorical latent variable $z$ is designed by:

$$
\begin{aligned}
&w^{(1)}, \cdots, w^{(K)}, \boldsymbol{d} = \mathrm{DNN}_z(\boldsymbol{x}; \theta_1), \\
&z \sim \mathcal{Categorical}(w^{(1)}, \cdots, w^{(K)}), \\
&\boldsymbol{h} = \boldsymbol{A}\mathrm{onehot}(z) + \boldsymbol{d}, \\
&\boldsymbol{\mu}_c, \boldsymbol{s}_c = \mathrm{DNN}_c(\boldsymbol{z}; \theta_2), \\
&\boldsymbol{c} \sim \mathcal{Logistic}(\boldsymbol{\mu}_c, \boldsymbol{s}_c), \\
&Pr(\boldsymbol{c} \preceq \boldsymbol{r}) = \prod_i \mathrm{sigmoid}\left(\frac{\boldsymbol{r}^{(i)} - \boldsymbol{\mu}_c^{(i)}}{\boldsymbol{s}_c^{(i)}}\right).
\end{aligned}
\tag{25}
$$

Then we can estimate the probability of $y$ by:

$$
p_\theta(y|\boldsymbol{x}, \boldsymbol{r}) = \sum_{k=1}^K p_{\theta_1}(z = k|\boldsymbol{x}) p_{\theta_2}(y|z = k, \boldsymbol{r}).
\tag{26}
$$

In the categorical case, we can easily consider all possible values of $z$, therefore we do not need to introduce the recognition model which provides a better distribution for stochastic sampling.

## B.3  GCM for Continuous Regression

When $y$ is a continuous variable, we can transform the regression problem into a binary classification problem according to Section 4.1. Here we demonstrate how to obtain the maximum likelihood estimate.

First, we build the generative model for $t$ and $c$, such that

$$
Pr(y + t > 0|\boldsymbol{z}) = Pr(\boldsymbol{r} \succ \boldsymbol{c}|\boldsymbol{z}).
\tag{27}
$$

We suppose $y$ is a Gaussian variable, i.e. $y|z \sim \mathcal{N}(\mu, \sigma^2)$, where $\sigma = F_\sigma(z)$ is a learnable variable and $\mu$ needs to be solved according to Equation 27. Since we have

$$
\Phi\left(\frac{\mu + t}{\sigma}\right) = Pr(\boldsymbol{r} \succ \boldsymbol{c}|\boldsymbol{z}) \triangleq p_1,
\tag{28}
$$

then we can solve $\mu$ as

$$\hat{\mu} = \sigma \Phi^{-1}(p_1) - t, \tag{29}$$

which is also the maximum likelihood estimation of $y$. The loss of GCM-VI can be formulated as:

$$\mathcal{L} = \frac{(y - \hat{\mu})^2}{2\sigma^2} + \log \sigma - \log \frac{p_\theta(\boldsymbol{z}|\boldsymbol{x})}{q_\phi(\boldsymbol{z}|\boldsymbol{x}, \boldsymbol{r}, y)}, \tag{30}$$

where $\boldsymbol{z} \sim q_\phi(\boldsymbol{z}|\boldsymbol{x}, \boldsymbol{r}, y)$. So we can now train our model and estimate $y$.

## C  ABLATION STUDIES

### C.1  ABLATION ON LATENT DIMENSION AND SAMPLE NUMBER

We perform ablation studies for the GCM-VI method based on the Adult dataset, evaluating three main hyperparameters: $D$, the latent dimension and $K$, the sampling number. We take $D$ and $K$ from $\{2, 4, 8, 16, 32\}$ separately and repeat the experiment 8 times with different random seeds, and here is the result.

Table 5: Experimental results (ACC) on the Adult dataset with multiple $D$ and $K$ settings.

|          | $D = 2$          | $D = 4$          | $D = 8$          | $D = 16$         | $D = 32$         |
|----------|------------------|------------------|------------------|------------------|------------------|
| $K = 2$  | 0.8858 ±0.0016   | 0.8855 ±0.0016   | 0.8855 ±0.0017   | 0.8852 ±0.0015   | 0.8847 ±0.0018   |
| $K = 4$  | 0.8857 ±0.0018   | 0.8853 ±0.0017   | 0.8852 ±0.0014   | 0.8850 ±0.0015   | 0.8849 ±0.0016   |
| $K = 8$  | 0.8858 ±0.0017   | 0.8858 ±0.0019   | 0.8855 ±0.0016   | 0.8852 ±0.0017   | 0.8852 ±0.0015   |
| $K = 16$ | 0.8857 ±0.0017   | 0.8861 ±0.0016   | 0.8854 ±0.0013   | 0.8848 ±0.0017   | 0.8854 ±0.0015   |
| $K = 32$ | 0.8856 ±0.0013   | 0.8855 ±0.0013   | 0.8857 ±0.0015   | 0.8853 ±0.0014   | 0.8853 ±0.0014   |

We can see that for low-dimensional revenue and cost variables, taking $D$ and $K$ small is sufficient to generate $\boldsymbol{c}$.

### C.2  ABLATION ON TYPE OF LATENT VARIABLE

We compare the categorical and Gaussian settings of the latent variable $\boldsymbol{z}$. Here is the result:

Table 6: Experimental results of GCRM on the multiple datasets.

| Method              | Adult
ACC↑     | COMPAS
ACC↑    | Diabetes
ACC↑  | Blog Feedback
RMSE↓ |
|---------------------|-------------------|-------------------|-------------------|------------------------|
| GCM (Categorical)   | 0.8850 ±0.0013    | 0.6983 ±0.0010    | **0.8443** ±0.0003 | **0.0988** ±0.0010     |
| GCM (Gaussian)      | 0.8854 ±0.0013    | 0.6991 ±0.0011    | 0.8441 ±0.0001    | 0.0994 ±0.0003         |
| GCM VI (Gaussian)   | **0.8858** ±0.0014 | **0.7011** ±0.0011 | 0.8442 ±0.0002    | 0.1005 ±0.0004         |

We can see that GCM-VI and GCM-categorical perform the best, this is consistent with their objectives, since GCM-categorical is trained by the exact $LL$ and GCM-VI provides a better estimation of the latent $\boldsymbol{z}$ than the original GCM.

# D  Cogeneration of Cost and Revenue

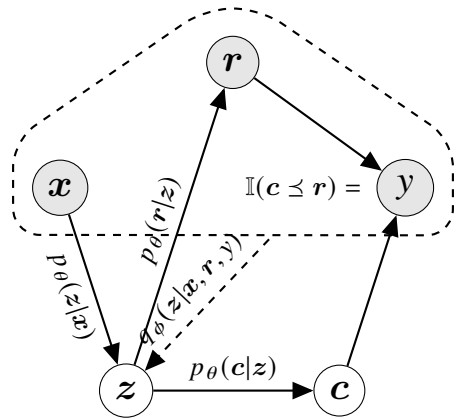

Figure 5: The generative graph for $p(y, r | x, z)$.

In certain cases, the assumption of conditional independence $z \perp\!\!\!\perp r \mid x$ may be too restrictive. Instead, we can adjust the cost generative model $p(c|x)$ to a cost-revenue generative model $p(c, r|x)$, as illustrated in Figure 5. In this context, we establish another weaker conditional independence relationship: $x \perp\!\!\!\perp r \mid z$. Similar to Equation 16, the ELB is given by:

$$
\log p_\theta(y, r | x)
$$
$$
\geq \mathbb{E}_{z_k \sim q_\phi} \log \left[ \frac{1}{K} \sum_{k=1}^{K} \frac{p_\theta(y, r, z | x)}{q_\phi(z | x, r, y)} \right] \tag{31}
$$
$$
= \mathbb{E}_{z_k \sim q_\phi} \log \left[ \frac{1}{K} \sum_{k=1}^{K} \frac{p_{\theta_3}(r|z) p_{\theta_2}(y|r, z, x) p_{\theta_1}(z|x)}{q_\phi(z|x, r, y)} \right].
$$

Here, the generation of $r$ follows the same procedure as generating $c$:

$$
\lambda_r = \text{DNN}_r(z; \theta_3), \quad p_{\theta_3}(r|z) = \mathcal{P}(r; \lambda_r). \tag{32}
$$

We perform experiments of the cogeneration of cost and revenue (noted as GCRM-VI) on multiple dataset, and the results are shown in Table 7. It shows that the effect of GCRM-VI is close to the original GCM-VI method. This shows optimistic potential for the cogeneration method for GCM.

Table 7: Experimental results of GCRM on the multiple datasets.

| Method | Adult
ACC↑ | COMPAS
ACC↑ | Diabetes
ACC↑ | Blog Feedback
RMSE↓ |
|---|---|---|---|---|
| GCM | 0.8854 ±0.0013 | 0.6991 ±0.0011 | 0.8441 ±0.0001 | **0.0994** ±0.0003 |
| GCM VI | **0.8858** ±0.0014 | **0.7011** ±0.0011 | **0.8442** ±0.0002 | 0.1005 ±0.0004 |
| GCRM VI | **0.8858** ±0.0011 | 0.6985 ±0.0018 | 0.8438 ±0.0003 | 0.1025 ±0.0032 |

# E  Experimental Details

## E.1  Detailed Results

The details of our experiments on the four public datasets are shown in the following tables.

Table 8: Detailed result of experiments on the Adult dataset.

| Method | Log Loss | RMSE | AUC | ACC |
|---|---|---|---|---|
| MLP | 0.2352 ±0.0030 | 0.2578 ±0.0017 | 0.7836 ±0.0057 | 0.8837 ±0.0012 |
| MM | 0.2355 ±0.0029 | 0.2578 ±0.0018 | 0.7827 ±0.0052 | 0.8836 ±0.0010 |
| SMM | 0.2351 ±0.0027 | 0.2577 ±0.0017 | 0.7833 ±0.0051 | 0.8837 ±0.0011 |
| CMNN | 0.2379 ±0.0027 | 0.2588 ±0.0016 | 0.7780 ±0.0053 | 0.8832 ±0.0013 |
| Hint | 0.2661 ±0.0027 | 0.2660 ±0.0018 | 0.7829 ±0.0058 | 0.8846 ±0.0011 |
| PWL | 0.2352 ±0.0028 | 0.2578 ±0.0017 | 0.7833 ±0.0055 | 0.8835 ±0.0012 |
| GCM | 0.2321 ±0.0030 | 0.2569 ±0.0017 | 0.7934 ±0.0054 | 0.8854 ±0.0013 |
| GCM VI | **0.2315** ±0.0030 | **0.2568** ±0.0017 | **0.7948** ±0.0049 | **0.8858** ±0.0014 |

Table 9: Detailed result of experiments on the COMPAS dataset.

| Method | Log Loss | RMSE | AUC | ACC |
|---|---|---|---|---|
| MLP | 0.5951 ±0.0014 | 0.4516 ±0.0006 | 0.7427 ±0.0010 | 0.6955 ±0.0008 |
| MM | 0.5925 ±0.0010 | 0.4504 ±0.0005 | 0.7450 ±0.0007 | 0.6949 ±0.0021 |
| SMM | 0.5925 ±0.0005 | 0.4504 ±0.0002 | 0.7447 ±0.0006 | 0.6955 ±0.0020 |
| CMNN | 0.5951 ±0.0013 | 0.4515 ±0.0006 | 0.7441 ±0.0008 | 0.6997 ±0.0011 |
| Hint | 0.6055 ±0.0010 | 0.4567 ±0.0005 | 0.7343 ±0.0012 | 0.6861 ±0.0024 |
| PWL | 0.5947 ±0.0014 | 0.4515 ±0.0006 | 0.7429 ±0.0012 | 0.6960 ±0.0013 |
| GCM | 0.5922 ±0.0007 | 0.4501 ±0.0004 | 0.7461 ±0.0008 | 0.6991 ±0.0011 |
| GCM VI | **0.5913** ±0.0008 | **0.4498** ±0.0004 | **0.7472** ±0.0007 | **0.7011** ±0.0011 |

Table 10: Detailed result of experiments on the Diabetes dataset.

| Method | Log Loss | RMSE | AUC | ACC |
|---|---|---|---|---|
| MLP | 0.3130 ±0.0002 | 0.3114 ±0.0001 | 0.8250 ±0.0004 | 0.8431 ±0.0004 |
| MM | 0.3153 ±0.0008 | 0.3125 ±0.0004 | 0.8211 ±0.0013 | 0.8409 ±0.0008 |
| SMM | 0.3159 ±0.0016 | 0.3128 ±0.0008 | 0.8200 ±0.0025 | 0.8401 ±0.0013 |
| CMNN | 0.3176 ±0.0017 | 0.3137 ±0.0008 | 0.8174 ±0.0028 | 0.8393 ±0.0015 |
| Hint | 0.3808 ±0.0044 | 0.3370 ±0.0017 | 0.8144 ±0.0008 | 0.8407 ±0.0005 |
| PWL | 0.3144 ±0.0002 | 0.3121 ±0.0001 | 0.8227 ±0.0003 | 0.8417 ±0.0003 |
| GCM | **0.3128** ±0.0001 | 0.3112 ±0.0001 | **0.8253** ±0.0001 | 0.8441 ±0.0001 |
| GCM VI | 0.3129 ±0.0001 | **0.3112** ±0.0000 | 0.8252 ±0.0002 | **0.8442** ±0.0002 |

Table 11: Detailed result of experiments on the Blog Feedback dataset.

| Method | MSE Loss | RMSE |
|---|---|---|
| MLP | 0.0109 ±0.0001 | 0.1042 ±0.0004 |
| MM | 0.0121 ±0.0004 | 0.1100 ±0.0018 |
| SMM | 0.0124 ±0.0002 | 0.1114 ±0.0008 |
| CMNN | 0.0125 ±0.0001 | 0.1118 ±0.0005 |
| Hint | 0.0125 ±0.0003 | 0.1118 ±0.0013 |
| PWL | 0.0114 ±0.0001 | 0.1069 ±0.0006 |
| GCM | **0.0099** ±0.0001 | **0.0994** ±0.0003 |
| GCM VI | 0.0101 ±0.0001 | 0.1005 ±0.0004 |

# F COMPARISON OF TIME COMPLEXITY

One of the key advantages of our GCM model is its efficiency during the inference stage. For each given $x$, the model can easily calculate $p_\theta(y|x, r_i)$ for multiple $r_i$ values. This efficiency arises because the GCM model predicts the latent variables $z$ and $c$ based solely on $x$, allowing it to subsequently predict $y$ using $c$ and $r_i$. As a result, we avoid the computation of inputting each pair of $(x, r_i)$ into a deep neural network as methods. We evaluated the inference efficiency for various numbers of $r$ while keeping $x$ stable, and the results are presented in Table 12. As demonstrated, the GCM becomes the fastest method when the number of $r$ exceeds 64, validating its inference efficiency in multi-revenue prediction scenarios. When the number of $r$ reaches the extreme value of 1024, GCM can save up to 72% time cost compared to the fastest baseline model.

Table 12: Inference time cost (ms per batch) of different models with different numbers of $r$ on the COMPAS dataset.

| Method | Inference $r$ numbers per given $x$ | | | | | | | | | | |
|---|---|---|---|---|---|---|---|---|---|---|---|
| | 1 | 2 | 4 | 8 | 16 | 32 | 64 | 128 | 256 | 512 | 1024 |
| MM | 1.51 | 2.35 | 3.33 | 4.83 | 9.27 | 17.36 | 31.24 | 58.53 | 112.65 | 306.57 | 308.33 |
| CMNN | 3.39 | 5.17 | 9.02 | 15.87 | 28.95 | 51.96 | 102.01 | 198.07 | 394.63 | 869.76 | 877.47 |
| PWL | **1.02** | **1.67** | **2.47** | **3.73** | **7.86** | **13.89** | 26.01 | 47.86 | 92.95 | 280.70 | 285.48 |
| GCM | 11.66 | 11.55 | 11.98 | 12.89 | 13.88 | 16.85 | **20.14** | **28.89** | **43.88** | **76.23** | **79.63** |