# OpenReview forum: "Reformulating Strict Monotonic Probabilities with a Generative Cost Model"
_ICLR.cc/2025/Conference — Submitted to ICLR 2025_

### Official Review · Reviewer_CJwc · 2024-10-31

**Soundness:** 2
**Presentation:** 2
**Contribution:** 3
**Rating:** 5
**Confidence:** 4

**Summary:**

The paper proposes a latent variable model capturing both a monotonic part and a part that need not be monotonic. They give some background, and then propose a loss function for a variational training with neural nets.

**Strengths:**

It is an interesting problem and their construction is quite novel and creative.

**Weaknesses:**

The objective function and implementation leaves a lot of details missing and unexplained. The paper does not inspire confidence in the method.

**Questions:**

Overview with some questions embedded:

Lines 054-059 could be clearer with a graphical model diagram. Edit: there is one later, nice.

Line 063 may not be quite right; p_theta(x,r) cannot be ignored in the evidence unless you drop the dependence on theta. Maybe rework the setup / explanation a bit, this shouldn’t be a big issue.

Line 101 equivalent to what?

Line 150 consider the machinery within https://arxiv.org/abs/2301.11695 and also the separate line of work on normalizing flows that all involve e.g. invertibility, monotone transformations, etc.

Line 210 (0,1) should be {0,1}

Line 189 should **y** and r have the same dimension?

Line 289 elementwise

Line 304 looks like it would suffer from high variance since pi(z) is fixed but p(z|x) depends on x.

Line 323 the combination of losses looks a little suspicious.

Line 323 Note that IWAE (Burda) is equal to ELBO (Jordan et al) for IWAE number of samples set to one; why is adding ELBO to IWAE sensible? Shouldn’t pi be something else and then no ELBO? Please expand, this is unconvincing.

Line 324 it seems like this doesn’t really match the beta vae setup; it would if you put a beta in front of the kl in line 318. What is this doing?

Line 378 how about some ablations of alpha and beta terms? Edit: there are some in appendix C.

Table 1 does not seem to match appendix C for any value of the parameters, what is missing? Please add details.

Line >= 378 how did you set all of the parameters?

Line 698 affect

---

> ### Author Response · Authors · 2024-11-21
>
> Thank you for the comprehensive review and valuable suggestions. We have revised our paper as you requested and here are our replies (updated in 2024-11-29).
>
> \# Question 1: $p(x,z)$ is ignored in the evidence.
>
> We ignore $p(x,r)$ because we always have the given variables $x$ and $r$, which are sampled from their prior $p(x, r)$. So, sampling $(x,r,y)$ from $p(x,y,r)$ requires only sampling $y$ based on $p(y|x,r)$ by the given $(x,r)$.
>
> \# Question 2: Should $y$ and $r$ have the same dimension?
>
> No, it only requires $y|(x,r_1) \prec y|(x,r_2)$ for any $r_1 \prec r_2$. The definition of monotonic conditional probability is in the background section (Section 2 - Monotonic Conditional Probability).
>
> \# Question 3: The high variance issue for importance sampling.
>
> We modified the model for $z$ being a categorical variable. In the original version, we assume that $z$ is a $K$-categorical variable that $z \in \{1,\cdots,K\}$, and we take the sampling distribution $\pi(z)\equiv \frac{1}{K}$, this leads to the evidence estimator by importance sampling as $p(y|z,r)\approx \frac{1}{N}\sum_{j=1}^{N}p(z_k|x)p(y|z_k,r)$, where $z_k \sim \pi$. This is equivalent to the law of total probability when we take $K=N$ and $z_k=k$. As a result, we are aware that we do not need to introduce the importance sampling technique. In the new revision, we change the model for categorical latent variable based on the law of total probability. For complex latent distribution assumptions, we follow the reparameterization trick as in VAE and use the recognition model $q_\phi(z|x,r,y)$ to sample $z$, which helps reduce the high variance of the evidence estimator.
>
> \# Question 4: The combination of losses.
>
> Thanks for the suggestion, we rethink about this and realized that the combination of LL and ELB is not really necessary. We also remove the KL divergence term of $D_{KL}(p(z|x)\|p(z))$ in $\mathcal L_{GCM}$, and now the loss functions of GCM and GCM-VI become:
>
> $\mathcal L_{GCM}(\theta; x,  r, y)= -\log \left[\frac{1}{K}\sum_{k=1}^{K}p_{\theta_2}\left(y|{x},{r},{z}_k\right)\right]\approx -LL,$
>
> where $z_k \sim p_{\theta_1}(z|x)$, and
>
> $\mathcal L_{GCM-VI}(\theta, \phi; x,  r, y)=  -\log \left[\frac{1}{K}\sum_{k=1}^K\frac{p_{\theta_2}(y| z_k,  r)p_{\theta_1}(z_k| x)}{q_{\phi}(z_k| x,  r, y)}\right]=-ELB\geq -LL,$
>
> where $z_k \sim q_\phi(z|x,r,y)$.
>
> Therefore, in the new formulation, we do not need the combination $\mathcal L=-LL-\alpha ELB + \beta KL(p(z|x)\|p(z))$ and the hyperparameters of $\alpha$ and $\beta$ are dropped. So we also remove the ablation study on $\alpha$ and $\beta$, and add an ablation study on the sample number $K$ and latent dimension $D$ in Appendix E.
>
> The experimental result of this new setting still shows that GCM-Vi performs best in 3 of the four public datasets. Proving that the combination of $LL$ and $ELB$ is not necessary.
>
> \# Question 5: About IWAE and ELBO.
>
> In the latest revision, we follow the form of original IWAE for the GCM-VI loss. This is shown in our reply to question 4, the experimental results are also renewed.
>
> \# Question 6: The loss does not match $\beta$ -VAE.
>
> To further simplify the model, we remove the additional KL divergence term $D_{KL}(p_\theta(z|x) \| p(z))$. As a result, the $\beta$ parameter is removed from the latest revision.
>
> \# Question 7: Table 4 (table 1 in the original paper) does not match the ablation study.
>
> This is because we used $\alpha=0.5$ in the experiment section, but did not include it in the ablation study (where we take $\alpha \in \{ 0, 0.2, 0.4, \cdots, 2.0 \}$) and we also use different random seeds in the experiment part and the ablation part. So in the latest revision, the result of the experiment in Table 3 (Adult ACC =$0.8858\pm 0.0014$) is not identical to the ablation result in Table 5 (Adult ACC =$0.8855\pm 0.0013$ when $D=4$ and $K=32$).
>
> \# Question 8: Parameter details.
>
> We add the hyperparameter details in Section 5 (line 480), which we take $D=4$ and $K=32$ for all experiments on the public datasets, and we upload the code to: https://github.com/iclr-2025-4464/GCM, it has the full realization of the GCM, GCM-VI and baseline models, as well as the parameter settings.

---

### Official Review · Reviewer_e3QF · 2024-11-03

**Soundness:** 3
**Presentation:** 3
**Contribution:** 3
**Rating:** 6
**Confidence:** 2

**Summary:**

The paper proposes the Generative Cost Model (GCM) to enforce strict monotonicity by modeling a latent cost variable, using variational inference and importance sampling. This approach avoids architectural constraints and outperforms traditional methods on synthetic and real datasets.

**Strengths:**

The paper seems to present a novel approach to formulate the problem of strict monotonic probability, it provides convincing theoretical analysis and emprical results.

**Weaknesses:**

The paper lacks empirical validation across diverse, high-dimensional real-world datasets, limiting the demonstrated generalizability of the Generative Cost Model (GCM). Moreover, the code was not provided in the supplementary materials to assess the reproducibility of the results.

**Questions:**

1. Can the authors provide additional justification or empirical validation for the assumption of conditional independence between the latent variable z and revenue r given x, as this assumption may affect model flexibility in real-world applications?

2. Could the authors elaborate on the computational efficiency of the Generative Cost Model (GCM) compared to traditional monotonic models, especially when applied to high-dimensional datasets? How does the computational time compare to the benchmarks?

---

> ### Author Response · Authors · 2024-11-21
>
> Thank you for the reviews and valuable suggestions. We have revised our paper as you requested and here are our replies (updated in 2024-11-29).
>
> \# Weakness 1: Lack of validation.
>
> We added two more real-world datasets: Diabetes and Blog Feedback, where the Diabetes is a large dataset with 253,680 examples, and the task of Blog Feedback dataset is a regression task with 8 monotonic features and 272 nonmonotonic features. The experiment on both datasets proving our GCM method is consistently well performed. You can find the details in Table 3 in the experiment section and more detailed results in the tables in Appendix E.
>
> \# Weakness 2: Lack of code.
>
> We uploaded the anonymous code to https://github.com/iclr-2025-4464/GCM.
>
> \# Question 1: The assumption of  $ z\perp  r \mid  x$ needs additional justification.
>
> We rethink about this assumption and find that it is not necessary. In Appendix D, we build another generation model that generates both the revenue variable $r$ and the cost variable $c$ by the latent variable $z$, this helps our model avoid $ z\perp  r \mid  x$. We call the new cogeneration method the GCRM (generative cost-revenue model). We also performed an experiment based on GCRM, and found that this approach performs close to the original GCM in four public datasets, which shows an optimistic potential.
>
> \# Question 2: Elaborate on the computational efficiency.
>
> We validate the time consumption of different monotonic modeling methods, the detailed data are shown in Appendix F. The inference task requires one to estimate $p(y|x,r)$, where $x$ is fixed and $r$ varies in a set of candidates $\{r_1 , \cdots, r_n\}$. The GCM is slow when the candidate number $n$ is small, however, when the number $n$ grows, the GCM exceeds the baseline methods. The efficiency of GCM in the large number of candidates $r$ is that GCM does not need to put all pairs of $(x, r_i)$ in a deep neural network as traditional methods, which extend the computation time by $n$ times. Instead, GCM only requires us to put $x$ into a deep neural network once to generate $c$, which does not take additional time spent, after obtaining $c|x$, it is efficient to calculate $p(c \prec r_i)$ for $r_i \in \{r_1,…,r_n\}$ since this step does not need a DNN, we can simply solve it using equation (10). As a result, when the candidate number of $ r$ reaches the extreme value of 1024, GCM can save up to 72\% time cost compared to the fastest baseline model.

---

### Official Review · Reviewer_ta4m · 2024-11-03

**Soundness:** 3
**Presentation:** 3
**Contribution:** 3
**Rating:** 6
**Confidence:** 3

**Summary:**

This paper proposes a novel approach to model strict monotonic probabilities. Without loss of generality, it considers the binary classification formulation $y|x, r\sim Bernoulli(y;G(x,r))$, where $G(x,r)$ is a function that is monotonic in $r$. The target is to learn the function $G$. Instead of directly learning $G$, the paper introduces a cost variable $c$ and reformulates $G(x,r)$ as an integration $\int_{c<r} p(c|x)d c$. The paper then introduces a generative cost model to approximate the conditional distribution $p(c|x)$.

**Strengths:**

The paper tackles an interesting problem of modeling monotonic probabilities. The problem itself is important, and the reformulation proposed by the paper is unique. Extensive experiments are conducted to support this new method.

**Weaknesses:**

The paper lacks theoretical results on the finite sample efficiency of the proposed algorithm. For the experiment design, an important setup that requires strict monotonicity is quantile regression, where the conditional quantile should be monotonic in the quantile argument. It would be interesting to see experiments designed for it and a comparison with the existing benchmarks.

**Questions:**

No additional questions.

---

> ### Author Response · Authors · 2024-11-21
>
> Thank you for the reviews and valuable suggestions. We have revised our paper as you requested and here are our replies (updated in 2024-11-29).
>
> \# Weakness 1: The paper lacks theoretical results on the finite sample efficiency of the proposed algorithm.
>
> We find some difficulties in proving the finite sample efficiency. To prove the finite sample efficiency, we need to validate if the variance of an estimator $T$ of the parameter $\theta$ reaches the Cramér–Rao lower bound, i.e. ${\rm var} (T) = \mathcal I_\theta^{-1}$ holds. However, we do not have an analytical estimator $T(x,y,r)$ of the parameters $\theta$ and $\phi$, since they are optimized using gradient descent algorithms. Furthermore, we are unable to compute Fisher information $\mathcal I_\theta$, since it is computationally impossible to integrate a complex deep neural network.
>
> If we misunderstand your question, please inform us, and we will try to consider a proper explanation as soon as possible.
>
> \# Weakness 2: Application in quantile regression.
>
> It is an interesting application of monotonicity. We added an additional experiment of quantile regression through a simulation, and we put the details in Section 5.1. We show a comparison of the quantile curve between all methods in Figure 3 and MAE of the $r$th quantile predicted by all methods with different settings of $r$. The test result shows that GCM performs the best in MAE metrics. We can also see that the traditional strict monotonic methods (MM, SMM, CMNN) suffer from approximation accuracy, as the strict monotonic structures (e.g. positive weighting matrices and monotonic activations) weaken the universal approximation ability of neural networks. For the non-monotonic (MLP) and weak-monotonic (Hint, PWL) methods, they have better approximation accuracy than the strict monotonic methods, but their $r$th quantile curves of different $r$'s are not sufficiently separated as shown in Figure 3, due to the lack of strict monotonic constraints.

---

### Official Review · Reviewer_NjCz · 2024-11-08

**Soundness:** 3
**Presentation:** 3
**Contribution:** 2
**Rating:** 3
**Confidence:** 3

**Summary:**

The paper models a conditional monotonic distribution as a marginalization over latent variables. The key idea is to modify the monotonic modeling problem into modeling an element-wise cumulative distribution function (over a cost variable C). To actually model the latter, the authors introduce a latent variable modeling problem and solve it via a standard importance-weighted likelihood estimate with or without an additional ELBO term. The paper presents improved results in two experiments over a variety of baselines.

**Strengths:**

- Clean reformulation of the monotonic problem into a classification over a latent variable.
- Well-motivated techniques to solve the problem.
- Comparison against a variety of baselines.

**Weaknesses:**

No confidence intervals in the results tables. It's not clear that the improvement achieved by GCM is large enough (even over the simple MLP) to justify the much more complicated modeling procedure.

**Questions:**

- How wide are the confidence intervals for each of the metrics? Just do standard bootstrap if possible and report please.
- Why introduce the extra prior $p_\theta(Z)?$ Also, why is the prior different from $\pi(Z)$?

**Details Of Ethics Concerns:**

-

---

> ### Author Response · Authors · 2024-11-21
>
> Thank you for the reviews and valuable suggestions. We have revised our paper as you requested and here are our replies (updated in 2024-11-29).
>
> \# Weakness 1 \& question 1: No confidence intervals in the results tables.
>
> We add a 95\% confidence interval to the experimental results; the confidence intervals are estimated by repeating the experiments 10 times. The results with confidence interval can be found in Table 1 \& 3 in the section 5, and tables in Appendix A, C and E.
>
> \# Weakness 2: It's not clear that the improvement achieved by GCM is large enough.
>
> With the demonstration of confidence intervals, the improvement in the GCM compared to the baselines, including the MLP, is significant. The detailed experimental results are shown in Table 1 \& 3 in the section 5 and in Appendix E. As past studies (e.g. paper 1\&2 ) on monotonic networks show, the improvement of AUC and ACC on Adult and COMPAS datasets are not numerically large, due to the limitation (dataset size, random noise, etc.) of datasets. In fact, a 0.003 improvement in the AUC metric is considered huge in various scenarios.
>
> We also add two more sets of experiments on the Diabetes dataset and the Blog Feedback dataset, and the GCM-VI is still the best method in all datasets, and the results are reported in Appendix E.
>
> Moreover, MLP is not a monotonic model, we use it to see the effect if there is no monotonic constraint of a neural network. Due to the difficulty of training a strict monotonic neural network (for example, all the weighting matrix must be positive in the classic Min-Max monotonic network, see paper 3; can not apply the layer normalization technique, which is not monotonic; lack of scalability, see paper 4, etc), it is an achievement that a monotonic model ties with the unconstrained MLP, while preserving the advantage of giving strict monotonic predictions. In fact, not all monotonic models can defeat MLP due to the rigorous monotonic constraints, which are shown in Table 1 \& 3. However, in all sets of experiments, the GCM model surpasses the MLP and baseline models as shown in Table 1 \& 3, which proves the ability of our model.
>
> paper 1: https://arxiv.org/pdf/2306.01147
>
> paper 2: https://arxiv.org/pdf/2011.10219
>
> paper 3: https://proceedings.neurips.cc/paper/1997/file/83adc9225e4deb67d7ce42d58fe5157c-Paper.pdf
>
> paper 4: https://openreview.net/pdf?id=DjIsNDEOYX
>
> \# Question 2: Why introduce the extra prior $p_\theta(z)$? Also, why is the prior different from $\pi(z)$?
>
> We take the sampling distribution $\pi(z)$ same as the prior $p(z)$ in the original paper. In the latest revision, we remove the $\pi(z)$ and modify the evidence estimator from $p_{\theta}(y|x, r)\approx \frac{1}{k}\sum_{j=1}^{k}\frac{p_{\theta_1}(z_j|x)}{\pi( z_j)}p_{\theta_2}(y|z_j, r)$ to the exact estimator $p_{\theta}(y|x, r)=\sum_{j=1}^{k}p_{\theta_1}(z_j=k|x)p_{\theta_2}(y|z_j=k, r)$ for a categorical variable $z$, which is equivalent to the importance sampling when $z$ is categorical and $\pi(z)\equiv {\rm constant}$. In the new revision, we also provide an estimator of the evidence for a Gaussian latent variable $z$ using the reparameterization trick instead of importance sampling, where $z_j=z(\theta_1, \epsilon_j) \sim p_{\theta_1}(z|x)$ and $p_{\theta}(y|x, r)\approx \frac{1}{K}\sum_{j=1}^{k}p_{\theta_2}(y|z_j, r)$, avoiding the high-variance issue in importance sampling. The experimental results in Section 5.1 are updated using the Gaussian latent variable $z$. We also compared the performance between Gaussian and categorical latent variables in Appendix C.2, showing that they can achieve similar performance in four public datasets.

---

### Author Response · Authors · 2024-11-29
**Key points of the final revision (part I)**

Dear all reviewers, thank you for all the comprehensive reviews and valuable suggestions, we have submitted the final revision, and here are the modifications we made comparing to the original paper.

## 1. We revised the latent distribution and simplified the loss functions of GCM and GCM-VI.

The latent distribution we used in the original paper is categorical, and to deal with it, we have to use importance sampling to estimate evidence. However, if we traverse all possible values of $z$, it is equivalent to the law of total probability, which gives the exact estimate of the evidence by:

$p_\theta(y|z,r)= \frac{1}{K}\sum_{j=1}^{K}p_{\theta_1}(z=k|x)p_{\theta_2}(y|z=k,r).$

In this case, we can directly optimize the evidence and we have the loss function:

$\mathcal L_{GCMcate}(\theta; x,  r, y)= -\log \left[\frac{1}{K}\sum_{j=1}^{K}p_{\theta_1}(z=k|x)p_{\theta_2}(y|z=k,r)\right].$

For complex latent variables, for example, $z$ form the Gaussian distribution, we cannot use the law of total probability (impossible to integrate on $z$) or importance sampling (high variance). We follow the classical reparameterization trick and reformulate the loss of GCM as:

$\mathcal L_{GCM}(\theta; x,  r, y)= -\log \left[\frac{1}{K}\sum_{k=1}^{K}p_{\theta_2}\left(y|{x},{r},{z}_k\right)\right],$

where $z_k \sim p_{\theta_1}(z|x)$. To make a better estimate of $z$, we adopt a recognition model $q_\phi(z|x,r,y)$, similar to the IWAE, the variational version of GCM (GCM-VI) has a loss function formulated as:

$\mathcal L_{GCM-VI}(\theta, \phi; x,  r, y)=  -\log \left[\frac{1}{K}\sum_{k=1}^K\frac{p_{\theta_2}(y| z_k,  r)p_{\theta_1}(z_k| x)}{q_{\phi}(z_k| x,  r, y)}\right],$

where $z_k \sim q_{\phi}(z|x,r,y)$. The three revised loss functions are simplified compared to the original paper. We remove the hyperparameters $\alpha$ and $\beta$ in the original version formulated as $\mathcal L=-LL-\alpha ELB+\beta D_{KL}(p_\theta(z|x)\|p(z))$. So we now have fewer hyperparameters and the loss function is cleared without a redundant linear combination of $LL$ and $ELB$.

We performed an ablation study on these three loss functions in Appendix C.2, and the results are as follows.

|Method|	Adult ACC|	COMPAS ACC|	Diabetes ACC|	BlogFeedback RMSE|
|---------------------|----------------------------|----------------------------|----------------------------|----------------------------|
|GCM-Categorical|0.8850$\ \pm$0.0013| 0.6983$\ \pm$0.0010| 0.8443$\ \pm$0.0003| 0.0988$\ \pm$0.0010|
|GCM-Gaussian|0.8854$\ \pm$0.0013| 0.6991$\ \pm$0.0011| 0.8441$\ \pm$0.0001| 0.0994$\ \pm$0.0003|
|GCM-Gaussian-VI|0.8858$\ \pm$0.0014| 0.7011$\ \pm$0.0011| 0.8442$\ \pm$0.0002| 0.1005$\ \pm$0.0004|

We can see that GCM-VI and GCM-categorical perform the best, this is consistent with their objectives, since GCM-categorical is trained by the exact $LL$ and GCM-VI provides a better estimation of the latent $ z$ than the original GCM.

---

> ### Author Response · Authors · 2024-11-29
> **Key points of the final revision (part II)**
>
> ## 2. We replaced the card-gambling experiments with the quantile regression experiment.
>
> The original rules of the card-gambling experiment are hard to fully understand. Therefore, we adopt a simpler task of quantile regression based on simulations in Section 5.1. The task is formulated as follows:
>
> $${\rm sample}\ \ r \sim \mathcal U([0,1]), $$
>
> $${\rm sample}\ \ \hat y_r \sim p_\theta(y|x,r), $$
>
> $${\rm minimize} \ \ r(y - \hat y_r )_{+} +(1-r)(\hat y_r - y ) _{+}. $$
>
> Here $\hat y_r$ is the prediction of the $r$th quantile of $y|x$ and the objective function in the third line follows the classical quantile regression. According to the definition of the $r$th quantile, which is monotonic with respect to $r$, the prediction value $\hat y_r$ should also be monotonic with respect to $r$. So we can test the monotonic methods based on this task. In our experiment, the training examples of $x$ and $y$ are generated by:
>
> $${\rm sample}\ \ x \sim \mathcal U([-1.5,1.5]), $$
>
> $${\rm sample}\ \ \epsilon \sim \mathcal U([0, 1]), $$
>
> $$ y= 0.3 \sin(2 (x + 0.8)) + 0.4 \sin(3 (x - 1.3)) + 0.3 \sin(5 x) + 0.4 (0.8 x^2+0.6) \epsilon.$$
>
> The test results are shown in Figure 3 and Table 1 in Section 5.1. Proving that GCM has the best performance among all methods. We show the MAE metrics as follows.
>
> |Method| $r=0.1$|	$r=0.3$|	$r=0.5$|	$r=0.7$|	$r=0.9$|
> |-----------------|----------------------|----------------------|----------------------|----------------------|----------------------|
> |MLP|	   0.1495$\ \pm$0.0340|	0.1157$\ \pm$0.0283|	0.1057$\ \pm$0.0255|	0.1230$\ \pm$0.0309|	0.1477$\ \pm$0.0386|
> |MM|     0.2002$\ \pm$0.0572|	0.1103$\ \pm$0.0320|	0.0723$\ \pm$0.0245|	0.1067$\ \pm$0.0346|	0.1745$\ \pm$0.0495|
> |SMM|    0.2345$\ \pm$0.0693|	0.1194$\ \pm$0.0356|	0.0812$\ \pm$0.0246|	0.1236$\ \pm$0.0366|	0.1919$\ \pm$0.0556|
> |CMNN|   0.1768$\ \pm$0.0340|	0.1119$\ \pm$0.0174|	0.0823$\ \pm$0.0161|	0.1007$\ \pm$0.0198|	0.1480$\ \pm$0.0332|
> |Hint|   0.1402$\ \pm$0.0285|	0.1137$\ \pm$0.0263|	0.1068$\ \pm$0.0292|	0.1154$\ \pm$0.0368|	0.1316$\ \pm$0.0374|
> |PWL|    0.1793$\ \pm$0.0282|	0.1476$\ \pm$0.0164|	0.1394$\ \pm$0.0193|	0.1524$\ \pm$0.0216|	0.1698$\ \pm$0.0207|
> |GCM|	   0.0984$\ \pm$0.0188|	0.0777$\ \pm$0.0119|	0.0669$\ \pm$0.0096|	0.0759$\ \pm$0.0127|	0.0991$\ \pm$0.0211|
>
> ## 3. We renewed the experiments in public datasets.
>
> We added two public datasets, so now we have four public datasets in the experiment, they are:
>
> |dataset		  | total examples 	| dimension of x       |dimension of r 				| target
> |-----------------|----------------------|----------------------|----------------------|----------------------|
> |Adult			  | 48,842			| 33 				       |	4 						| classification
> |COMPAS			  | 7,214			| 9 					       |4 							| classification
> |Diabetes		  | 253,680			| 105 				       |4 							| classification
> |BlogFeedback	  | 52,397			| 272 				       |8 							| regression
>
> They are tested in Section 5.2, and GCM-VI achieve the best performance in Adult, COMPAS and Diabetes, while GCM performs the best in BlogFeedback.
>
> |Method|	Adult ACC|	COMPAS ACC|	Diabetes ACC|	BlogFeedback RMSE|
> |---------------------|----------------------------|----------------------------|----------------------------|----------------------------|
> |MLP| 	0.8837$\ \pm$0.0012|	0.6955$\ \pm$0.0008|	0.8431$\ \pm$0.0004|	0.1042$\ \pm$0.0004|
> |MM|    	0.8836$\ \pm$0.0010|	0.6949$\ \pm$0.0021|	0.8409$\ \pm$0.0008|	0.1100$\ \pm$0.0018|
> |SMM|   	0.8837$\ \pm$0.0011|	0.6955$\ \pm$0.0020|	0.8401$\ \pm$0.0013|	0.1114$\ \pm$0.0008|
> |CMNN|   0.8832$\ \pm$0.0013|	0.6997$\ \pm$0.0011|	0.8393$\ \pm$0.0015|	0.1118$\ \pm$0.0005|
> |Hint| 	0.8846$\ \pm$0.0011|	0.6861$\ \pm$0.0024|	0.8407$\ \pm$0.0005|	0.1118$\ \pm$0.0013|
> |PWL| 	0.8835$\ \pm$0.0012|	0.6960$\ \pm$0.0013|	0.8417$\ \pm$0.0003|	0.1069$\ \pm$0.0006|
> |GCM|   	0.8854$\ \pm$0.0013|	0.6991$\ \pm$0.0011|	0.8441$\ \pm$0.0001|	 0.0994$\ \pm$0.0003|
> |GCM VI|	0.8858$\ \pm$0.0014|	0.7011$\ \pm$0.0011|	 0.8442$\ \pm$0.0002|	0.1005$\ \pm$0.0004|
>
> ## 4. We uploaded the code.
>
> Our code is uploaded to <https://github.com/iclr-2025-4464/GCM>, welcome to try it and feel free to give us some advice.
>
> In the end, we sincerely appreciate your reviews and are eager to hear from you in the future.

---

### Meta-Review · Area_Chair_e65W · 2024-12-22

**Metareview:**

The paper addresses the problem of conditional monotonic distribution modeling by representing it as a marginalization over latent variables. The central idea is to reformulate the monotonic modeling challenge into modeling an element-wise cumulative distribution function (CDF) with respect to a cost variable CCC. To achieve this, the authors introduce a latent variable modeling framework, which is solved using a standard importance-weighted likelihood estimation, optionally incorporating an ELBO term. The original paper has many issues and errors. For example, the basic conditional independence assumption was found unnecessary in the rebuttal period. These major changes are not fully verified in the rebuttal phase and it was hard for the reviewers to fully review the new version again. I do not think the current version is ready for acceptance.

**Additional Comments On Reviewer Discussion:**

None of the reviewers engaged in discussions with the authors, likely because many of the concerns raised by the reviewers stem from errors made by the authors. While the authors attempted to address these issues in the revision, the extent of the changes is too substantial for a second review. Additionally, the significance of the results remains unconvincing. For instance, the authors argue that a 0.003 improvement in the AUC metric is highly significant in various scenarios, a claim that appears questionable.

---

### Decision · Program_Chairs · 2025-01-22

Reject